# Overfeeding during Lactation in Rats is Associated with Cardiovascular Insulin Resistance in the Short-Term

**DOI:** 10.3390/nu12020549

**Published:** 2020-02-20

**Authors:** Daniel González-Hedström, Lucía Guerra-Menéndez, Antonio Tejera-Muñoz, Sara Amor, María de la Fuente-Fernández, Beatriz Martín-Carro, Riansares Arriazu, Ángel Luis García-Villalón, Miriam Granado

**Affiliations:** 1Departamento de Fisiología, Facultad de Medicina, Universidad Autónoma de Madrid, 28029 Madrid, Spain; dgonzalez@pharmactive.eu (D.G.-H.); antoniotemu@gmail.com (A.T.-M.); sara.amor@uam.es (S.A.); maria.delafuente@uam.es (M.d.l.F.-F.); beatriz.martinc@uam.es (B.M.-C.); angeluis.villalon@uam.es (Á.L.G.-V.); 2Departamento de Ciencias Médicas Básicas, Universidad San Pablo CEU, 28003 Madrid, Spain; lguerra@ceu.es (L.G.-M.); arriazun@ceu.es (R.A.); 3CIBER Fisiopatología de la Obesidad y Nutrición. Instituto de Salud Carlos III, 28029 Madrid, Spain

**Keywords:** early overnutrition, childhood obesity, lactation, insulin, cardiovascular, rat

## Abstract

Childhood obesity is associated with metabolic and cardiovascular comorbidities. The development of these alterations may have its origin in early life stages such as the lactation period through metabolic programming. Insulin resistance is a common complication in obese patients and may be responsible for the cardiovascular alterations associated with this condition. This study analyzed the development of cardiovascular insulin resistance in a rat model of childhood overweight induced by overfeeding during the lactation period. On birth day, litters were divided into twelve (L12) or three pups per mother (L3). Overfed rats showed a lower increase in myocardial contractility in response to insulin perfusion and a reduced insulin-induced vasodilation, suggesting a state of cardiovascular insulin resistance. Vascular insulin resistance was due to decreased activation of phosphoinositide 3-kinase (PI3K)/Akt pathway, whereas cardiac insulin resistance was associated with mitogen-activated protein kinase (MAPK) hyperactivity. Early overfeeding was also associated with a proinflammatory and pro-oxidant state; endothelial dysfunction; decreased release of nitrites and nitrates; and decreased gene expression of insulin receptor (IR), glucose transporter-4 (GLUT-4), and endothelial nitric oxide synthase (eNOS) in response to insulin. In conclusion, overweight induced by lactational overnutrition in rat pups is associated with cardiovascular insulin resistance that could be related to the cardiovascular alterations associated with this condition.

## 1. Introduction

Obesity is a main health concern worldwide due to its association with metabolic syndrome and cardiovascular diseases [1]. Although it was previously considered that metabolic syndrome was rare in childhood, its incidence has markedly increased together with childhood obesity [2,3]. Indeed, type 2 diabetes and insulin resistance are some of the most prevalent alterations of metabolic syndrome among children [2,4]. 

In addition to metabolic disorders, obese children also show cardiovascular alterations [5], such as arterial hypertension, whose incidence in obese children or adolescents is around 20%–25% [6]. Furthermore, it is reported that childhood obesity is associated with morphological and functional vascular alterations, such as endothelial dysfunction, increased arterial stiffness, and coronary calcification [7]. In the heart, childhood obesity is correlated with left ventricular hypertrophy [7,8] and reduced systolic and diastolic function [9]. These cardiovascular alterations may be related, at least in part, with alterations in cardiovascular insulin sensitivity, as insulin exerts important effects on the cardiovascular system in physiological conditions [10]. In the blood vessels, it promotes vasodilation mainly through the release of nitric oxide [11], and in the heart it exerts a positive inotropic effect [12]. 

Both the vasodilator and inotropic effects of insulin are mediated by the activation of the intracellular phosphoinositide 3-kinase (PI3K)/Akt pathway [13,14], whereas the mitogen-activated protein kinase (MAPK) pathway exerts opposite effects [15]. 

Cardiovascular insulin effects are reduced in states of insulin resistance and this may contribute to the development of hypertension and and other the cardiovascular complications [16]. In addition, it has been proposed that the PI3K/Akt and MAPK pathways are affected differently by insulin resistance, leading to an imbalance in the vasodilator and vasoconstrictor effects of insulin [17].

The litter reduction model is an accepted experimental model of early overnutrition in rodents that consists of reducing the number of pups in the litters during the lactation period. This early intervention modifies both milk availability [18] and milk composition [19], and results not only in increased body weight, adiposity, and plasma levels of leptin, insulin, and cholesterol [20], but also in cardiovascular alterations, such as decreased heart contractility [21], increased blood pressure [18]**,** and altered response to angiotensin II in both the heart [21,22] and the blood vessels [23]. 

Regarding insulin sensitivity, previous studies have shown that early postnatal overnutrition in rats is associated with insulin resistance, both peripherally [24] and centrally [25]. However, although it is reported that exposure to certain toxic substances during the early postnatal period results in a state of cardiac insulin resistance [26,27], the cardiovascular insulin sensitivity in response to early overfeeding, at least to our knowledge, has not yet been studied. 

Therefore, the aim of this work was to study if early overnutrition in rats results in a state of cardiovascular insulin resistance that may explain, at least in part, the cardiovascular alterations associated to this condition. For this purpose, we used the model of postnatal overfeeding by litter size reduction in rats, which is an accepted model of childhood obesity [20]. 

## 2. Methods

### 2.1. Animals

All the experiments were conducted with the approval of the Animal Care and Use Committee of the Community of Madrid (Spain).

Four six-month-old Sprague-Dawley rat dams fed ad libitum with a standard diet (Altromin, Lage, Germany) were used for these studies. On the day of birth, some litters were adjusted to twelve pups (6 males and 6 females) per mother (L12, lean) and other litters were adjusted to three male pups per mother (L3, overfed). The rationale behind this distribution is the assumption that reduced litters have increased milk availability, and therefore the animals become overweight at the end of the lactation period (postnatal day 21). At weaning, all animals were subjected to overnight fasting and sacrificed by decapitation after an intraperitoneal injection of sodium pentobarbital (100 mg/kg). Immediately after sacrifice organs were collected and weighed. The blood was collected in tubes containing EDTA and centrifuged at 3000 rpm for 15 minutes at 4 °C to obtain the plasma. 

### 2.2. Plasma Measurements

Glycaemia was measured just before sacrifice in a blood drop obtained by tail puncture with Glucocard^TM^ G+ meter and Glucocard^TM^ GSensor (Koji, Konan-Cho, Koka-Shi, Shiga, Japan).

In the plasma, the levels of insulin, leptin, adiponectin, total lipids, triglycerides, total cholesterol, Low Density Lipoprotein (LDL) cholesterol, and High Density Lipoprotein (HDL) cholesterol were analyzed. 

The plasma concentrations of total lipids, triglycerides, total cholesterol, and LDL and HDL cholesterol were measured by commercial kits from Spinreact S.A.U (Girona, Spain) following the manufacturer instructions. The analysis of insulin, leptin, and adiponectin levels was performed using commercial enzyme-linked immunosorbent assay (ELISA) kits from Millipore (Dramstadt, Germany). All samples were run in duplicate and within the same assay for all analyses. The intra- and inter-coefficients of variation for all analyses were < 10% and < 20%, respectively. The minimum detectable concentrations (ng/mL) were: insulin, 13.0; leptin, 4.2.

### 2.3. Experiments of Heart Perfusion: Langendorff

After sacrifice, hearts were immediately removed and mounted in the Langendorff perfusion system as previously described [28]. After a 30 min equilibration period with constant flow perfusion, increasing doses of insulin were added to the perfusion solution (10^−10^–10^−7^ M) in the presence or absence of the blocker of the PI3K/Akt pathway wortmanin (10^−6^ M), or in the presence or absence of the MAPK inhibitor SCH-772984 (3.10^−6^ M). Both preincubations were performed for 30 minutes before insulin administration. Insulin (#I0516) and wortmanin (#681675) were purchased from Sigma-Aldridh (St. Louis, MO, USA) and SCH-772984 from Cayman Chemical (#19166, Ann Arbor, MI, USA). The effects of insulin administration on coronary perfusion pressure, left intraventricular pressure, dp/dt as an index of myocardial contractility, and heart rate were recorded. Control hearts were perfused during the same time without adding insulin. 

Coronary perfusion pressure was measured through a lateral connection in the perfusion cannula and left ventricular pressure was measured using a latex balloon inflated to a diastolic pressure of 5–10 mmHg, both of which were connected to Statham transducers (Statham Instruments, Los Angeles, CA, USA). Left ventricular pressure was recorded and was used to calculate the first derivative of the left ventricular pressure curve (dP/dt) as an index of heart contractility and heart rate. These parameters were recorded on a computer using the PowerLab/8e data acquisition system (ADInstruments, Colorado Springs, CO, USA).

Finally, to study the effects of insulin administration in the gene or protein expression of different markers in the myocardium, hearts from both experimental groups were perfused and incubated in the presence or absence of insulin 10^−7^ M for 30 min. Afterwards, hearts were collected and stored at −80 °C. 

### 2.4. Experiments of Vascular Reactivity

For the vascular reactivity experiments, the aorta was carefully dissected, cut into 2 mm segments, and kept in cold isotonic saline solution. The assembly of the segments was performed as previously described [23]. The changes in isometric force were recorded using a PowerLab data acquisition system (ADInstruments, Colorado Springs, CO, USA). After applying an optimal passive tension of 1 g, vascular segments were allowed to equilibrate for 60–90 min. Afterwards, segments were stimulated with potassium chloride (KCl 100 mM, #1.04936, Sigma-Aldrich, St. Louis, MO, USA) to determine the contractility of smooth muscle. Segments that failed to contract at least 0.5 g to KCl were discarded. After equilibration, the segments were precontracted with 10^−7.5^. M phenylephrine (#P1250000, Sigma-Aldrich, St. Louis, MO, USA) to subsequently perform a cumulative dose–response curve in response to insulin (10^−11^–10^−6^ M), acetylcholine (10^−9^–10^−4^ M), and sodium nitroprusside (10^−9^–10^−4^ M) (#A6625 and # 71778, Sigma-Aldrich, St. Louis, MO, USA) . The relaxation in response to insulin was determined based on the percentage of the active tone achieved by the nitric oxide (NO) donor sodium nitroprusside (10^−5^ M).

To study the mechanism of the vasodilation in response to insulin, some segments were preincubated for 30 min with the inhibitor of the nitric oxide synthase Nω-Nitro-L-arginine methyl ester hydrochloride (L-NAME) (10^−4^ M) (#N5751, Sigma-Aldrich, St. Louis, MO, USA), the inhibitor of cyclooxygenase sodium meclofenamate (10^−5^ M) (#1377803, Sigma-Aldrich, St. Louis, MO, USA), the inhibitors of potassium channels apamin (10^−6^ M) and charydobdotoxin (10^−7^ M) (#1289 and #C7802, Sigma-Aldrich, St. Louis, MO, USA) or the blocker of the PI3K/Akt pathway wortmanin (10^−6^ M). For each dose–response curve, the logarithm of the concentration producing 50% of the maximal response (ED50) was calculated by geometric interpolation.

### 2.5. Incubation of Aorta Segments in Presence/Absence of Insulin (10^−7^ M)

First, 2 mm thoracic aorta segments were incubated in 6-well culture plates (3 segments/well) with 1.5ml of Dulbecco’s modified Eagle’s medium and Ham’s F12 medium (DMEM/F-12) with glutamine from Gibco (1:1 mix; Invitrogen, Carlsbad, CA, USA), supplemented with 100 U/mL penicillin and 100 μg/mL streptomycin (Invitrogen, Carlsbad, CA, USA) or DMEM/F-12+insulin (10^−7^ M) (Sigma-Aldrich, St. Louis, MO, USA) at 37 °C in a 95 % O_2_ and 5 % CO_2_ incubator. After 30 min of incubation, both the segments and the culture media were collected and stored at –80 °C for further analysis. 

### 2.6. Nitrite and Nitrate Determination in the Culture Medium

Nitrite and nitrate concentrations, as indicators of nitric oxide release, were measured in the culture medium after incubation of aorta segments from both L12 and L3 rats in the presence or absence of insulin by a modified Griess assay method, described by Miranda et al. 2001 [29]. Briefly, 100 μL of vanadium chloride (#208272, Sigma-Aldrich, St. Louis, MO, USA) was added to 100 μl of culture medium on a 96-well plate. Immediately after, the Griess reagent (1:1 mixture of 1 % sulfanilamide (Merck Millipore, Darmstadt, Germany), and 0.1 % naphthylethylenediamine dihydrochloride (#106237, Merck Millipore, Darmstadt, Germany)) was added to each well and incubated at 37 °C for 30 min. The absorbance was measured at 540 nm. Nitrite and nitrate concentrations were calculated using a NaNO_2_ (#237213, Sigma-Aldrich, St. Louis, MO, USA) standard curve and was expressed in µM.

### 2.7. RNA Extraction and Quantitative RT Real Time PCR

Total RNA was extracted from 100 mg of myocardial (ventricular) and arterial tissue according to the tri-reagent protocol [30]. Then, cDNA was synthesized from 2 µg of total RNA using a high capacity cDNA reverse transcription kit (Applied Biosystems, Foster City, CA, USA). Glucose transporter type 4 (GLUT-4 ,Rn00562597_m1), insulin receptor (IR ,Rn00690703_m1), endothelial nitric oxide synthase (eNOS ,Rn02132634_s1), tumor necrosis factor α (TNFα ,Rn01525859_g1), interleukin 6 (IL-6 ,Rn01489669_m1), interleukin 1β (IL-1β ,Rn00580432_m1), cyclooxygenase 2 (COX-2 ,Rn01483828_m1), inducible nitric oxide synthase (iNOS ,Rn00561646_m1), superoxide dismutase 1 (SOD-1 ,Rn00566938_m1), glutathione reductase (GSR ,Rn01482159_m1), glutathione peroxidase 3 (GPX-3 ,Rn00574703_m1), lipoxygenase (LO ,Rn00563172_m1), NADPH (Nicotinamide adenine dinucleotide phosphate) oxidase 1 (NOX-1 ,Rn00586652_m1), and NADPH oxidase 4 (NOX-4 ,Rn00585380_m1) mRNA levels were assessed in heart and aorta samples by quantitative real-time polymerase chain reaction (PCR) using assay-on-demand kits (Applied Biosystems, Foster City, CA, USA). All samples were run in duplicate. TaqMan Universal PCR Master Mix (Applied Biosystems, Foster City, CA, USA) was used for amplification according to the manufacturer’s protocol in a Step One machine (Applied Biosystems, Foster City, CA, USA). Values were normalized to the housekeeping gene 18S (Rn01428915). According to the manufacturer’s guidelines, the ∆∆CT method was used to determine relative expression levels [31] in relation to gene expression levels in samples from L12 rats. 

### 2.8. Protein Quantification by Western Blot

First, 100 mg of myocardial (ventricular) or arterial tissue was homogenized using radioimmunoprecipitation assay buffer (RIPA buffer). After centrifugation (12000 rpm, 4 °C, 20 min), supernatant was collected and total protein content was measured by the Bradford method (Sigma-Aldrich, St. Louis, MO, USA). In each assay, the same amount of protein was loaded in each well (100 μg). After electrophoresis using resolving acrylamide sodium dodecyl sulfate (SDS) gels (8–10%) (Bio-Rad, Hercules, CA, USA), proteins were transferred to polyvinylidine difluoride (PVDF) membranes (Bio-Rad, Hercules, CA, USA). Transfer efficiency was determined by Ponceau red dyeing (Sigma-Aldrich, St. Louis, MO, USA). Membranes were then blocked with Tris-buffered saline (TBS) containing 5% (w/v) non-fat dried milk and incubated with the appropriate primary antibody for Akt (1:1000; # 04-796, Merk Millipore, Dramstadt, Germany), p-Akt (Ser 473)(1:500; #9271, Cell Signaling Technology, Danvers, MA, USA), MAPK (ERK 1/2) (1:1000; # ABS44, Merck Millipore, Dramstadt Germany), p-MAPK (1:500; #9102, Cell Signaling Technology, Danvers, MA, USA), eNOS (1:750; # ab76198, abcam, Cambridge, United Kingdom), and phospho-eNOS (Ser1177)( (1:500; # 07-428-I, Merck Millipore, Dramstadt, Germany). Membranes were subsequently washed and incubated with the secondary antibody conjugated with peroxidase (1:2000; Pierce, Rockford, IL, USA). Peroxidase activity was visualized by chemiluminescence and quantified by densitometry using BioRad Molecular Imager ChemiDoc XRS System (Hércules, CA, USA). 

Afterwards, membranes were also incubated with GAPDH (1:500; Ambion life technologies, Waltham, Massachusetts, USA) in order to normalize each sample for gel-loading variability. For each sample, relative protein expression levels were calculated in relation to protein expression levels in samples from L12 rats. 

### 2.9. Detection of Glucose Transporter 4 (GLUT-4) in the Heart by Immunofluorescence 

Hearts were fixed in 4% paraformaldehyde diluted in PBS solution for 24 hours and embedded in paraffin using the automatic equipment (Leica TP 1020, Leica, Switzerland). Longitudinal sections (3 μm-thick, HM 325, Microm) were stained with hematoxylin-eosin.

For GLUT-4 detection by immunofluorescence, ventricular sections were incubated with a specific antibody for human glucose transporter 4 (GLUT 4, rabbit anti-human GLUT4, ab654, Abcam), diluted 1/500 in PBS for 1 h, rinsed extensively with PBS, and incubated with the corresponding secondary antibody (goat anti-rabbit, Alexa Fluor 488; Molecular Probes) diluted to 1:1000. Then, to determine the translocation of GLUT-4 to the plasmatic membrane of cardiomyocytes, wheat germ agglutinin (WGA, Alexa Fluor™ 594) diluted to 1:200 in PBS was added and incubated for 45 min at room temperature in the dark. Finally, samples were rinsed successively with PBS, distilled water, and ethanol, and mounted with a drop of VECTASHIELD Antifade Mounting Medium containing 4’,6-diamidino-2-phenylindole (DAPI) (H-12000 Vector Laboratories), which fluoresces when bound to DNA. Images were obtained and scanned using a Leica SP8 Confocal Microscopy System coupled to a DMi6000 inverted microscope. Images were acquired with a 63x magnification, 1.4 NA PlanApo Oil objective and an additional confocal zoom of 2. As a source of excitation for the green and red channels, a white light laser was used, adjusting the wavelengths to the absorption optics of the fluorescent molecules. For visualization of DAPI, a 405 nm laser was used. Ten cells in each slide were randomly selected per sample for analysis of the GLUT4 in the heart. Quantification was made using a macro program and ImageJ software.

### 2.10. Statistical Analysis

All data are represented as mean ± SEM. Differences between L3 and L12 rats variables were examined by Student’s t-test or by two-way ANOVA followed by Bonferroni’s test for data obtained from experiments performed in the presence or absence of insulin. Differences were considered significant when *p* < 0.05.

## 3. Results

### 3.1. Body and Organ Weight

At birth, body weight did not differ between rats raised in control and reduced litters (Table 1). However, L3 rats showed increased body weight at weaning (*p* < 0.001), as well as increased visceral (*p* < 0.001), subcutaneous (*p* < 0.001), brown (*p* < 0.01), and periaortic (*p* < 0.05) fat weights compared to L12 rats. Regarding muscle mass, both gastrocnemius and heart weights were also significantly increased in L3 rats compared to L12 (*p* < 0.01 and *p* < 0.05 respectively). 

### 3.2. Glycemia, Lipid Profile and Plasma Concentrations of Metabolic Hormones

Table 2 shows a significant increase of glucose and insulin plasma levels in L3 rats compared to L12 (*p* < 0.05 for both). Likewise, plasma concentrations of leptin (*p* < 0.01), adiponectin (*p* < 0.01), total lipids (*p* < 0.01), and total cholesterol (*p* < 0.05) were significantly higher in overfed rats compared to controls. On the contrary, postnatal overfeeding induced a significant reduction in the plasma levels of HDL cholesterol (*p* < 0.05). No changes were found in the plasma levels of triglycerides and LDL cholesterol between experimental groups. 

### 3.3. mRNA Levels of Insulin Receptor and Glucose Transporter 4 in the Myocardium and GLUT-4 Localization

The mRNA levels of insulin receptor and glucose transporter 4 are shown in Figure 1. Overfed rats showed an upregulation in the gene expression of both IR (*p* < 0.05; Figure 1A) and GLUT-4 (*p* < 0.05; Figure 1B) in the myocardium compared to control rats. However, quantification of GLUT-4 by immunofluorescence showed a reduced localization of GLUT-4 in the cell membrane of cardiomyocytes in hearts from overfed pups compared to controls (*p* < 0.001; Figure 1C,D) 

### 3.4. Changes in Heart Rate, Coronary Perfusion Pressure and Heart Contractility (dp/dt) in Response to Insulin Administration

Basal heart rate was 280 ± 16 and 297 ± 12 beats/min in L12 and L3, respectively, and it was not modified by insulin treatment (data not shown).

The changes in coronary perfusion pressure and heart contractility (dp/dt) in response to insulin administration are represented in Figure 2A,B, respectively.

Insulin administration to perfused hearts from L12 rats induced vasodilatation of coronary arteries at 10^−8^ and 10^−9^ M concentrations, and vasoconstriction at the highest concentration used (10^−7^ M), whereas in hearts from L3 rats only the vasoconstriction with 10^−7^ M concentration was observed.

Before insulin treatment, the hearts from rats raised in reduced litters showed decreased contractility compared to hearts from rats raised in control litters (1699 ± 7 vs. 2331 ± 6 mmHg/s; *p* < 0.05). Insulin administration to perfused hearts induced a significant increase in heart contractility, both in L12 and in L3 rats, with this increase being significantly lower in hearts from overweight rats at insulin concentrations of 10^−9^ and 10^−8^ M (*p* < 0.05 for both). Preincubation with the blocker of the PI3K/Akt pathway wortmanin blunted the increase in dp/dt in both control and in overfed hearts at all dosages studied. Finally, preincubation with the blocker of the MAPK pathway SCH-772984 before the insulin dose–response curve significantly attenuated the early overfeeding-induced decrease in heart contractility in response to insulin at 10^−10^, 10^−9^, and 10^−7^M (*p* < 0.05 for all).

### 3.5. Myocardial Activation of PI3K/Akt and MAPK Pathways in Response to Insulin Administration

The two-way ANOVA analysis revealed no interaction between factors for p-Akt (Figure 3A), total Akt (Figure 3B), p-Akt/Akt ratio (Figure 3C), p-MAPK (Figure 3D), and total MAPK (Figure 3D); and a significant effect of insulin in both types of litters for p-Akt, p-Akt/Akt ratio, and p-MAPK (*p* < 0.01 for all). However, there was significant interaction between the two factors for p-MAPK/MAPK ratio (Figure 3F; F = 11.60; *p* < 0.01) and a significant effect of insulin was found only in hearts from rats raised in reduced litters (*p* < 0.001). 

No changes between experimental groups were found in the myocardial levels of p-eNOS, eNOS, and the ratio p-eNOS/eNOS (data not shown).

### 3.6. Vascular Reactivity of Aortic Rings in Response to Acetylcholine (Ach) and Sodium Nitroprusside (NTP)

The vascular response of aortic rings from L12 and L3 rats to Ach and NTP are shown in Figure 4A,B, respectively. Aortic rings from overfed rats showed decreased vasorelaxation in response to high concentrations of Ach (10^−6^ and 10^−5^ M) (*p* < 0.05 for both). Likewise, vasorelaxation in response to NTP was significantly reduced in L3 rats compared to controls at dosages between 10^−8^ and 10^−6^ M (*p* < 0.05 for all).

### 3.7. Vascular Reactivity, Phospho-eNOS Expression and Nitrites Release of Aortic Rings in Response to Insulin 

Aortic segments from control rats showed increased vasodilation in response to accumulative concentrations of insulin (10^−11^–10^−5^ M) compared to aortic segments from overfed rats at the doses of 10^−7^ M (*p* < 0.05), 10^−6^ M (*p* < 0.01), and 10^−5^ M (*p* < 0.01) (Figure 5A).

Results of nitrite and nitrate release and eNOS activation in aorta segments from both lean and overfed rats in response to insulin are shown in Figure 5B,C, respectively. 

In both cases, we found no interaction between both factors. In response to insulin 10^−7^ M, nitrite and nitrate concentrations (µM) were significantly up-regulated in the culture medium of aortic rings from L12 rats (*p* < 0.01), but not in aortic rings from L3 rats (Figure 5B). Likewise, insulin significantly increased the levels of p-eNOS in the aorta from lean (*p* < 0.05) but not from overfed rats (Figure 5C) whereas the content of total eNOS was unchanged (Figure 5D)

### 3.8. Vascular Reactivity of Aortic Rings in Response to Insulin in Presence/Absence of Meclofenamate, Apamine/Charibdotoxine, L-NAME, or Wortmanin 

Figure 6 shows the vascular response of aortic segments from both control and overfed rats to insulin in presence or absence of meclofenamate (A), apamine or charibdotoxine (B), L-NAME (C), or wortmanin (D).

In all cases, no significant interaction was found between factors (litter size and treatment with blockers). As indicated by the Area Under the Curve (AUC) values, insulin-induced vasodilation of aortic rings was partially reduced in L12 rats in the presence of L-NAME (*p* < 0.01) and wortmanin (*p* < 0.01). However, insulin-induced vasodilation of aortic rings from L3 rats was only attenuated in the presence of L-NAME (*p* < 0.01).

### 3.9. Activation of PI3K/Akt and MAPK Pathways in Response to Insulin Administration in Arterial Tissue

The activation of PI3K/Akt and MAPK pathways in arterial tissue in the presence or absence of insulin is shown if Figure 7. 

In response to insulin 10^−7^ M, no interaction and no significant changes were found in the activation of MAPK pathway in either aorta segments from L12 or in aorta segments form L3 rats. On the contrary, insulin induced a significant activation of the PI3K/Akt pathway in arterial tissue from lean rats, as is shown by the significant interaction (F = 5.44, *p* < 0.05) and the increase in the arterial content of p-Akt (*p* < 0.05, Figure 7A) and the arterial p-Akt/Akt ratio (*p* < 0.01, Figure 7C) only in arterial segments from L12 rats.

### 3.10. Gene Expression of IR, GLUT-4, and eNOS in Arterial Tissue in Response to Insulin

Figure 8 shows the mRNA levels of IR (A), GLUT-4 (B), and eNOS (C) in aorta segments from L12 and L3 rats in the presence or absence of insulin. 

A significant interaction between factors was found for IR (F = 5.41; *p* < 0.05), GLUT-4 (F = 6.72; *p* < 0.05), and eNOS (F = 5.49; *p* < 0.05). The post-hoc analysis revealed that in basal conditions, there was a downregulation in the gene expression of IR (*p* < 0.001), GLUT-4 (*p* < 0.05), and eNOS (*p* < 0.05) in overweight rats compared to controls. Insulin administration significantly up-regulated the gene expression of IR (*p* < 0.01), GLUT-4 (*p* < 0.05) and eNOS (*p* < 0.01) in aorta segments from L12 but not from L3 rats. 

### 3.11. Gene Expression of Inflammatory and Oxidative Stress-Related Markers in Myocardial and Aortic Tissue

The gene expression of different inflammatory and oxidative stress-related markers in myocardial and aortic tissue are shown in Table 3 and Table 4, respectively.

In the myocardium, early overnutrition was associated with a significant increase in the gene expression of the proinflammatory markers IL-1β (*p* < 0.05), IL-6 (*p* < 0.01), TNF- α (*p* < 0.05), and LO (*p* < 0.05), and the pro-oxidant enzymes NOX-1 (*p* < 0.05) and NOX-4 (*p* < 0.01). However, the gene expression of the antioxidant enzymes SOD-1 and GSR was also significantly upregulated in myocardial tissue from L3 rats compared to L12 (*p* < 0.05 for both). 

In the aorta, no changes were found in most of the markers, except for a significant overexpression of IL-6 and iNOS, and a downregulation in the mRNA levels of the antioxidant enzyme SOD-1 (*p* < 0.05 for all).

## 4. Discussion

In this study we show that early overnutrition in rats is associated with cardiovascular insulin resistance, both in the heart and in the aorta, and that this state may be responsible, at least in part, for the development of the cardiovascular alterations associated to this condition.

Our results show that insulin induces vasodilation in aorta segments of rats from both experimental groups. Arterial vasodilatation in response to insulin has been described before and is mediated, at least in part, by the release of endothelial nitric oxide [32]. Nitric oxide also seems to mediate insulin-induced vasodilation of aorta segments in our rats, as the relaxation in response to insulin was significantly reduced by the nitric oxide inhibitor L-NAME and incubation with insulin significantly stimulated nitrite release in aorta segments from L12 rats. 

It is reported that both the metabolic and vascular effects of insulin are mainly mediated by activation of the PIK3/Akt pathway [13,14]. Likewise, in this study insulin-induced vasodilation and incubation of aorta segments with insulin increased the p-AKT/Akt ratio in aorta segments from lean rats, and insulin-induced vasodilatation was reduced in the presence of the inhibitor of the PIK3/Akt pathway wortmanin. 

Obesity is associated with a reduced effect of insulin on glucose cell uptake, and it has been described that it also impairs its vasodilating arterial effect, which may contribute to the development of metabolic syndrome [17,33]. The reduced vasodilator effect in response to insulin in overfed rats was also associated with a decreased gene expression of insulin receptor, Glut-4 transporter, and eNOS gene expression, which resulted in decreased NO production. These results clearly indicate an impairment of the intracellular mechanisms that insulin triggers, resulting in a state of vascular insulin resistance, as occurred in previous studies that reported a decreased activation of this pathway in states of insulin resistance, both in vivo [34,35] and in vitro [36]. The reduction of insulin-induced vasodilatation in overfed rats may be related to a general impairment of arterial vasodilatation during the first stages of weight gain, as both the relaxation in response to acetylcholine, which is mediated by nitric oxide release from the endothelium, and to sodium nitroprusside, which acts directly on the vascular smooth muscle, were reduced in early overfed rats. Likewise, it is reported that vasodilatation to reactive hyperemia correlates negatively with body fat in adolescents [37]. As previously reported [38,39], the vascular insulin resistance may be due to both increased vascular inflammation and oxidative stress, since both the mRNA levels of pro-inflammatory and pro-oxidant markers were upregulated in overfed rats. However, adiponectin levels were also increased in over-nourished rats, which may be related to the unchanged activation of the MAPK pathway in arterial tissue.

The insulin effect on the heart has been less studied than that in the arteries, but there are several studies reporting a positive inotropic effect [12]. As expected, our results also show a positive inotropic effect of insulin in the heart, as indicated by the increased intraventricular pressure and dP/dt in response to insulin in both experimental groups, which agrees with previous studies [40,41]. As previously described [42], this effect is also mediated by the activation of the PI3K/Akt pathway, as cardiac insulin administration significantly increases the p-Akt/Akt ratio in the myocardium, and this effect is blocked by wortmanin. The increased myocardial contractility in response to insulin was reduced in hearts of overweight rats, indicating that insulin resistance is also present in the cardiac function. This decreased cardiac insulin sensitivity in overfed rats could be related, at least in part, with the decreased heart contractility previously described in this experimental model of early overnutrition [21]. However, the decreased contractility seemed not to be related with morphological or functional myocardial alterations, as we did not find any changes in either the cardiomyocyte area or in the expression of contractile proteins between L12 and L3 rats. 

Our results also show that the mechanisms of insulin resistance may be different in the heart and in the arteries. In the myocardium, the increase in the p-Akt/Akt ratio induced by insulin was not different in either experimental groups. Moreover, the expression of insulin receptor and Glut-4 transporter was not reduced but increased in overweight rats, although its translocation to the cardiomyocyte cell membrane was significantly lower. Since the activation of the PI3K/Akt pathway in response to insulin was similar in hearts from control and overfed rats, these results suggest that this pathway may not be the one responsible for cardiac insulin resistance in this experimental model of early overnutrition. Thus, we explored the activation of the MAPK pathway in response to insulin and we observed that the ratio p-MAPK/MAPK was enhanced by insulin to a greater extent in overweight rats than in controls. It has been described that the activation of the MAPK pathway inhibits myocardial contraction [43,44] by enhancing dephosphorylation of alpha-tropomyosin [45], and is involved in the development of insulin resistance in the heart [46]. Our results support this idea, as preincubation of hearts with a MAPK blocker attenuates the early overfeeding-induced decrease in heart contractility in response to insulin. Therefore, the activation of this pathway in overfed rats may counteract the activation of the PI3K/Akt pathway and result in a reduced cardiac contractility in response to insulin. The hyperactivity of the MAPK pathway may be related to the pro-inflammatory state in the hearts of overweight rats, where we found an increased expression of proinflammatory cytokines. Likewise, MAPK activation drives inflammatory processes in cardiomyocytes after LPS stimulation [47], doxorubicin toxicity [48]**,** or diabetes [49]. Inflammation in obesity induces cardiomyocyte cell death [50] and is a predictor of cardiac disease in states of insulin resistance [51]. For this reason, MAPK inhibitors are being studied as possible cardioprotective agents [52]. 

Altogether, our results show that early postnatal overfeeding is associated with a state of cardiovascular insulin resistance that affects both the aorta and the myocardium. However, the intracellular mechanisms that result in this state seem to be different in the heart and in the vessels, with the decreased activation of the PI3K/Akt pathway being responsible for vascular resistance in the aorta and the overactivation of the MAPK pathway being responsible for cardiac insulin resistance in the myocardium. 

In conclusion, the results of the present study indicate that early overnutrition in rats is associated with decreased insulin sensitivity in the short-term, both in the heart and in the aorta. This suggests that in a context of increased nutrients and energy supply, alterations of insulin signaling in the vasculature and heart arise early, possibly explaining the finding of heart dysfunction in obese children [53,54,55,56] and highlighting the importance of an early intervention.

## Figures and Tables

**Figure 1 nutrients-12-00549-f001:**
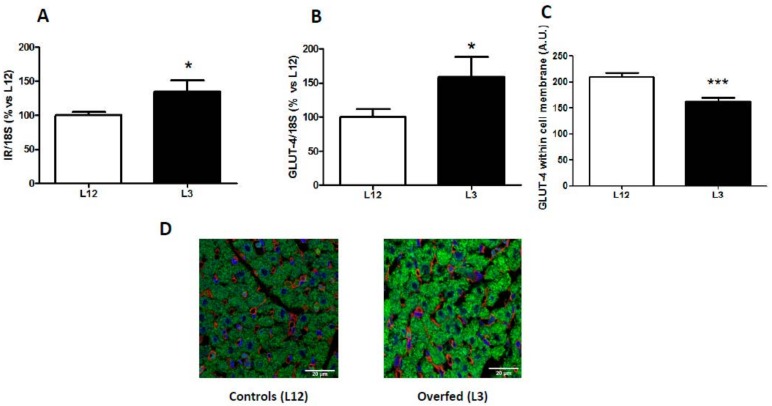
Gene expression of (**A**) insulin receptor (IR) and (**B**) glucose transporter 4 (GLUT-4), and GLUT-4 localization (**C**,**D**) in hearts from rats raised in L12 or L3 litters. Note: * *p* < 0.05 difference between L3 and L12; *** *p* < 0.001 difference between L3 and L12. Values are represented as mean ± SEM (*n* = 4–5 rats/experimental group) and expressed as % vs. L12. All samples were run in duplicate. Data were analyzed by Student’s *t*-test. Scale bar in D denotes 20 µm.

**Figure 2 nutrients-12-00549-f002:**
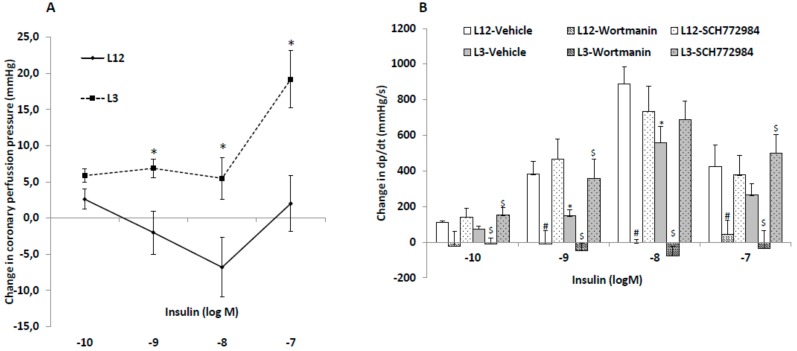
Changes in coronary perfusion pressure (**A**) and dP/dt (**B**) in response to insulin (10^−10^–10^−7^ M) compared to basal levels before insulin administration, in perfused hearts from rats raised in L12 or L3 litters, in the absence or presence of the phosphoinositide 3-kinase (PI3K) inhibitor wortmanin (10^−6^ M) or the inhibitor SCH-772984 (3.10^−6^ M). Note: * *p* < 0.05 difference between L3 and L12; ^#^
*p* < 0.05 difference between hearts in the presence or absence of wortmannin; ^$^
*p* < 0.05 difference between hearts in the presence or absence of SCH-772984. Values are represented as mean ± SEM; *n* = 6–9 rats/experimental group. Data were analyzed by Student’s *t*-test for each insulin dose.

**Figure 3 nutrients-12-00549-f003:**
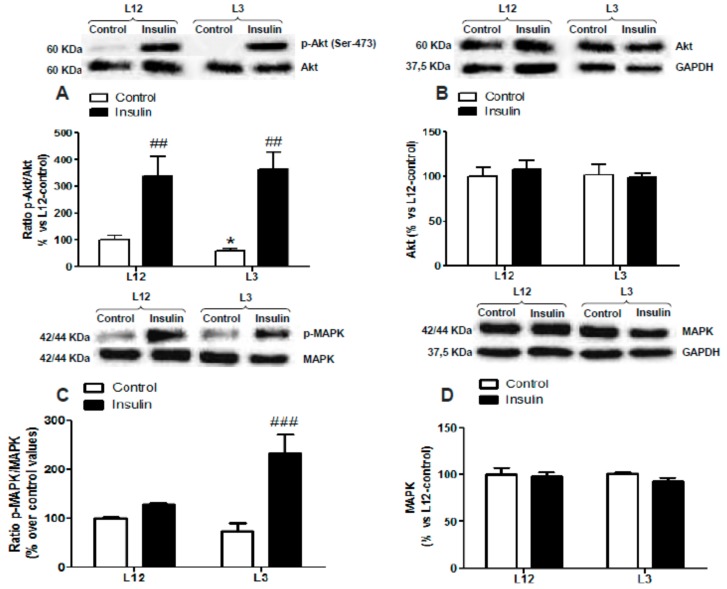
Protein levels of (**A**) ratio of phosphorylated to total Akt (p-Akt/Akt), (**B**) total Akt (Akt), (**C**) ratio of phosphorylated to total mitogen-activated protein kinase (MAPK) (p-MAPK/MAPK), and (**D**) total MAPK (MAPK) in perfused hearts from rats raised in L12 or L3 litters, incubated either with vehicle (control) or with insulin (10^−7^ M) for 30 min. Note: * *p* < 0.05 difference between control hearts from L3 and L12; ^##^
*p* < 0.01; ^###^
*p* < 0.001 difference between hearts incubated with insulin and control. Values are represented as mean ± SEM (*n* = 4–5 rats/experimental group) and expressed as % vs. L12. Data were analyzed by two-way ANOVA followed by Bonferroni post-hoc test.

**Figure 4 nutrients-12-00549-f004:**
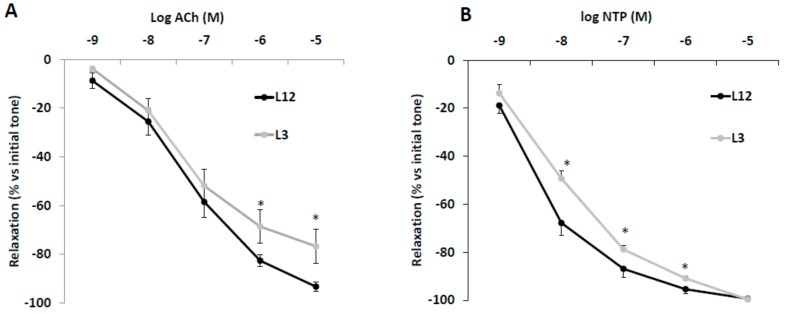
Relaxation caused by (**A**) acetylcholine (Ach, 10^−9^–10^−5^ M) or (**B**) sodium nitroprusside (NTP, 10^−9^–10^−5^ M) in precontracted aortic rings from rats raised in L12 or L3 litters. Note: * *p* < 0.05 difference between aortic rings from L3 and L12. Values are expressed as percentage of initial tone and represented as mean ± SEM (*n* = 9–12 rats/experimental group). Data were analyzed by Student’s *t*-test for each insulin dose.

**Figure 5 nutrients-12-00549-f005:**
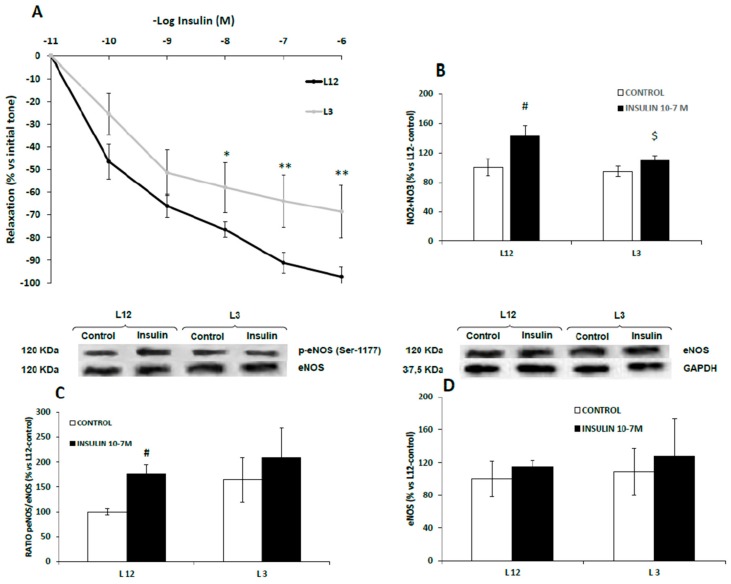
Relaxation to insulin (10^−11^–10^−6^ M) of precontracted aortic rings (**A**), nitrite and nitrate release (**B**), ratio of phosphorylated to total eNOS (p-eNOS/eNOS) (**C**) and total eNOS (eNOS) (**D**) in aorta segments from rats raised in L12 or L3 litters incubated over 30 days with either vehicle (control) or insulin (10^−7^ M). Note: * *p* < 0.05; ** *p* < 0.01 difference between aortic rings from L3 and L12; ^#^
*p* < 0.05; ^$^
*p* < 0.05 difference between hearts incubated with insulin from L3 and L12. Values are represented as mean ± SEM (*n* = 4–5 rats/experimental group) and expressed as % vs. L12. (*n* = 6–9 rats/experimental group). Data were analyzed by two-way ANOVA followed by Bonferroni post-hoc test.

**Figure 6 nutrients-12-00549-f006:**
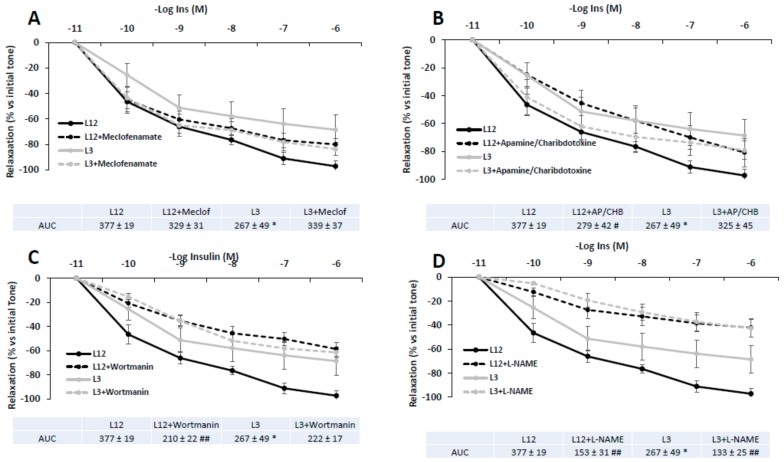
Relaxation in response to insulin (10^−11^–10^−6^ M) of precontracted aortic rings from rats raised in L12 or L3 litters in the absence or in the presence of **A**) the cyclooxygenase inhibitor meclofenamate (10^−5^ M), **B**) the potassium channels blockers apamin (10^−6^ M) plus charybdotoxin (10^−7^ M), **C**), the PI3K inhibitor wortmanin (10^−7^ M), and **D**), the nitric oxide synthase inhibitor L-NAME (10^−4^ M). Note: * *p* < 0.05 difference between aortic rings from L3 and L12; ^#^
*p* < 0.05; ^##^
*p* < 0.01 difference between aortic rings incubated with the blockers and control. Values are expressed as percentage of initial tone and represented as mean ± SEM; *n* = 6–13 rats/experimental group. Data were analyzed by two-way ANOVA followed by Bonferroni post-hoc test.

**Figure 7 nutrients-12-00549-f007:**
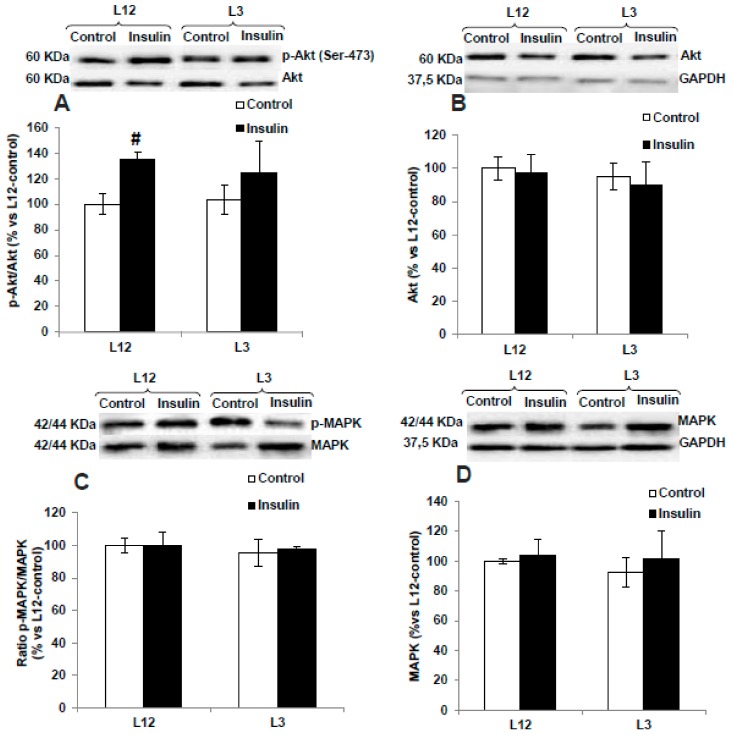
Protein levels of (**A**) ratio of phosphorylated to total Akt (p-Akt/Akt), (**B**) total Akt (Akt), (**C**) ratio of phosphorylated to total MAPK (p-MAPK/MAPK), and (**D**) total MAPK (MAPK) in aortas from rats raised in L12 or L3 litters, incubated with vehicle (control) or with insulin (10^−7^ M) for 30 min. Note: ^#^
*p* < 0.05; difference between hearts incubated with insulin and control. Values are represented as mean ± SEM (*n* = 6–9 rats/experimental group) and expressed as % vs. L12. Data were analyzed by two-way ANOVA followed by Bonferroni post-hoc test.

**Figure 8 nutrients-12-00549-f008:**
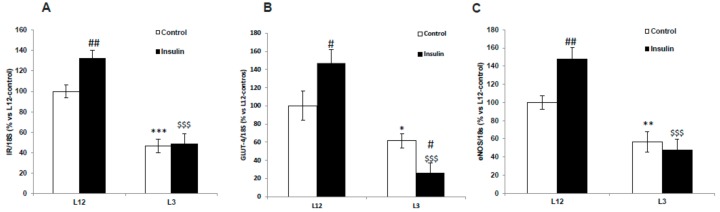
Gene expression of (**A**) insulin receptor (IR), (**B**) glucose transporter 4 (GLUT-4), and (**C**) endothelial nitric oxide synthase (eNOS) in aortic rings incubated for 30 days with vehicle (control) or insulin (10-7 M), from rats raised in L12 or L3 litters. Note: * *p* < 0.05; ** *p* < 0.01 *** *p* < 0.001 difference between aortic rings from L3 and L12; ^#^
*p* < 0.05; ^##^
*p* < 0.01 difference between hearts incubated with insulin and control; ^$$$^
*p* < 0.001 difference between aortas incubated with insulin from L3 and L12. Values are represented as mean ± SEM (*n* = 6–9 rats/experimental group) and expressed as % vs. L12. All samples were run in duplicate. Data were analyzed by two-way ANOVA followed by Bonferroni post-hoc test.

**Table 1 nutrients-12-00549-t001:** Body and organ weights from L12 (lean) and L3 (overfed) rats.

	L12	L3
**Body weight at birth (g)**	6.8 ± 1.1	6.9 ± 1.1
**Body weight at weaning (g)**	50 ± 1.1	65.7 ± 1.1 ***
**Visceral Epididymal adipose tissue (mg)**	99 ± 9,4	185 ± 11 ***
**Subcutaneous lumbar adipose tissue (mg)**	245 ± 26.7	521 ± 40.6 ***
**Brown adipose tissue (mg)**	259 ± 9	344 ± 19.3 ***
**Periaortic adipose tissue (mg)**	13.6 ± 1.2	21.1 ± 1.7 **
**Gastrocnemius (mg)**	161 ± 15	218 ± 12.2 **
**Heart (mg)**	336± 34.5	400 ± 21.7 *

Data are represented as mean ± SEM; *n* = 12–15 rats/group; * *p* < 0.05 vs. L12; ** *p* < 0.01 vs. L12. *** *p* < 0.001 vs. L12.

**Table 2 nutrients-12-00549-t002:** Glycemia and plasma levels of insulin, leptin, adiponectin, triglycerides, total cholesterol, LDL cholesterol, and HDL cholesterol from L12 (lean) and L3 (overfed) rats.

	L12	L3
**Glycemia (mg/dL)**	95 ± 13	107 ± 2.5 **
**Insulin (ng/mL)**	4.8 ± 0.8	14.1 ± 2.9 *
**Leptin (ng/mL)**	5 ± 0.5	17.3 ± 3.4 **
**Adiponectin (µg/mL)**	76 ± 7.1	117.5 ± 10.5 **
**Total Lipids (mg/dL)**	300 ± 16	375 ± 21 **
**Triglycerides (mg/dL)**	175.8 ± 20.4	171.3 ± 24.4
**Total Cholesterol (mg/dL)**	255.2 ± 13.2	292.1 ± 13.1*
**LDL-Cholesterol (mg/dL)**	153.4 ± 10.8	151.6 ± 9.1
**HDL- Cholesterol (mg/dL)**	137.6 ± 10.6	112.9 ± 6.6 *

Data are represented as mean ± SEM; n = 12–15 rats/group; * *p* < 0.05 vs. L12; ** *p* < 0.01 vs. L12.

**Table 3 nutrients-12-00549-t003:** Gene expression of interleukin 1-beta (IL-1β), interleukin- 6 (IL-6), tumor necrosis factor alpha (TNF-α), inducible nitric oxide synthase (iNOS), cyclooxygenase- 2 (COX-2), NADPH oxidase 1 (NOX-1), NADPH oxidase 4 (NOX-4), superoxide dismutase 1 (SOD-1), glutathione reductase (GSR), glutathione peroxidase 3 (GPX-3), and lipoxygenase (LO) in myocardial tissue from L12 (lean) and L3 (overfed) rats.

	L12	L3
**IL-1β**	100 ± 6.8	128 ± 12.3 *
**IL-6**	101 ± 6.2	134 ± 4.6 **
**TNF-α**	100 ± 6.1	149 ± 31.8 *
**iNOS**	100 ± 38.3	244 ± 141
**COX-2**	100 ± 32.8	42 ± 13.7
**NOX-1**	100 ± 24	235 ± 50 *
**NOX-4**	100 ± 15	209 ± 16 **
**SOD-1**	100 ± 5.6	139 ± 17.3 *
**GSR**	100 ± 11	143 ± 19 *
**GPX-3**	100 ± 8.2	115 ± 10
**LO**	100 ± 15.8	210 ± 68 *

Data are represented as mean ± SEM and expressed as % vs. L12. *n* = 6–8 samples/group; * *p* < 0.05vs. L12. ** *p* < 0.01 vs. L12. All samples were run in duplicate.

**Table 4 nutrients-12-00549-t004:** Gene expression of interleukin 1-beta (IL-1β), interleukin- 6 (IL-6), tumor necrosis factor alpha (TNF-α), inducible nitric oxide synthase (iNOS), cyclooxygenase- 2 (COX-2), NADPH oxidase 1 (NOX-1), NADPH oxidase 4 (NOX-4), superoxide dismutase 1 (SOD-1), glutathione reductase (GSR), glutathione peroxidase 3 (GPX-3), and Lipoxygenase (LO) in aortic tissue from L12 (lean) and L3 (overfed) rats.

	L12	L3
**IL-1β**	100 ± 23	219 ± 65
**IL-6**	101 ± 6	114 ± 2.5 *
**TNF-α**	100 ± 12	111 ± 11
**iNOS**	100 ± 19	1053 ± 362 *
**COX-2**	100 ± 23	211 ± 81
**NOX-1**	100 ± 8	127 ± 12 *
**NOX-4**	100 ± 5	99 ± 4
**SOD-1**	100 ± 9	72 ± 11 *
**GSR**	100 ± 5	93 ± 3
**GPX-3**	100 ± 6	89 ± 6
**LO**	100 ± 16	92 ± 10

Data are represented as mean ± SEM and expressed as % vs. L12. *n* = 6–9 samples/group; * *p* < 0.05 vs. L12. All samples were run in duplicate.

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
