# Peer review of "Overfeeding during Lactation in Rats is Associated with Cardiovascular Insulin Resistance in the Short-Term"

_nutrients, 2020, doi:10.3390/nu12020549_

Round 1

Reviewer 1 Report

Comments to the Author

In this manuscript, the authors investigate the role of overfeeding during the lactation period (rat model) and its effect on cardiovascular insulin resistance at PND21. This study concludes that overweight induced by early (lactational) overnutrition in rats is associated with cardiovascular insulin resistance. Though there are few exciting outcomes in this study, there are some suggestions and concerns, which must be addressed.

Major comments

Must the title be modified technically in rats’ starts to wean after PND21, when those rats sacrificed? It is suggestive of removing the words “at weaning” (if authors sacrificed early PND between 22-23; and mention the day of sacrifice in method section). The abstract conclusion needs to be rewritten: Line: 24 & 27. “…..by early overnutrition in rats” may be rewritten as “lactational overnutrition in rat pups is associated with cardiovascular …..”. Instead of early overnutrition, it is suggestive to use lactational. Same for instead of rat either pups or offspring. Introduction and methods: Lines 55-61. The rationale for the model is quite impressive, but from a translational point of view, human produces one offspring unlike rats; how a rat lactational overnutrition model “adjusting pups” comparable to childhood obesity in human; overfeeding starts after lactation in human? Provide Justification. Lines71-81: Male: female culling ratio or littermate adjustments must be provided. Lines71-81: Suckling effect increases more milk production/secretion technically L12, whereas L3 less milk production; did any study showed higher nutritional content in less litter adjusted compared to normal? Reference or justifiable must be provided. Which portion of myocardium used in this study for immunostaining and blotting and PCR? Figure 1D. Feeble image “it does not seem like 63x image”; provide better high-resolution clear image and scale bar must be provided with microphotograph and in legend mention magnification and scale bar values. Table 1: those organ weights are wet weight? Heart weight increased in L3 rats is that due to hypertrophy or hyperplasia or any other changes? Brown adipose also increased in L3, provide discussion for that results. Table 2: along with leptin, adiponectin also increased; in general, adiponectin favors insulin sensitivity, provide discussion. Provide all the primer/probe details used in this study, Gene expression table3 and 4; few primers details missing in methods, Check. IR has two isoforms, which one author measured, or else this probe or primer for both. Western blots: phosphorylated forms must be normalized with respective proteins (e.g., p-Akt/Akt; not with GAPDH) and unphosphorylated or total protein of interest normalized with an internal control (e.g., Akt/GAPDH); it must be chanced uniformly and accordingly change throughout the manuscript for all immunoblots.  Besides, mention the phosphorylated sites in blots (p-Akt Ser 473.) When writing results say, for example, in the line 306, “the expression of p-eNOS…” it may be rewritten as “the level of p-eNOS”; it is a posttranslational modification. Discussion: line 375, “…..for the first time…” maybe modified as “as per our knowledge, there is no study.” There are a couple of recent studies showing short-term and long term cardiac insulin resistance development during postnatal/lactational insults with toxicity, and those references (PMID: 24297258 and PMID: 30362281) may be included in discussion portions.

Other:

Must provide correct company name, place, and catalog for all chemicals and assay kits used. For ELISA kits provide inter-assay and intra-assay CVs. Especially, Western blot Ab Cat# must any other. Figure 7. Provide representative blots, see GAPDH for Akt or p-Akt faint band compared to GAPDH for p-MAPK, Why this difference? Did authors load the same amount of proteins in all the experimental conditions? QRT-PCR, what was the technical and biological replicates used, must be given in methods, as well as in figure legends. Abbreviate at first instance, check throughout the manuscript. There are many other minor errors of syntax and grammar throughout the text, which need to be fixed. Check manuscript preparation guidelines and modify them accordingly.

Reviewer 2 Report

In this manuscript, Daniel Gonzalez-Hedström et al. studied the possible association of lactation overfeeding in rats with the incidence of cardiovascular insulin resistance at weaning. The paper is well written clear and easy to read. The topic is hot and therefore, the fact that they point out that over lactation could lead to metabolic diseases can be problematic. The world organization advice to breastmilk until the six months of newborn life.

Says that scientific data are important and therefore in the experimental design please explain why 12 pups for rat if the nipples are 10. To reflect human behavior (usually 2 nipples for one kid) the control group should be thought 5 pups for one rat.

Author Response

In this manuscript, Daniel Gonzalez-Hedström et al. studied the possible association of lactation overfeeding in rats with the incidence of cardiovascular insulin resistance at weaning. The paper is well written clear and easy to read. The topic is hot and therefore, the fact that they point out that over lactation could lead to metabolic diseases can be problematic. The world organization advice to breastmilk until the six months of newborn life. Says that scientific data are important and therefore in the experimental design please explain why 12 pups for rat if the nipples are 10. To reflect human behavior (usually 2 nipples for one kid) the control group should be thought 5 pups for one rat

The objective of the reduced litter model is that L12 rats have a lower weight that L3 rats. To attain this, in this group the number of pups must be higher than the available nipples, so they have to compete for feeding and therefore the average amount of milk received by each is reduced. Rats from litters with 5 pups do not show significantly different body weight compared with 3-pup litters, so they cannot be used as controls to study the effects of increased weight gain.

Round 2

Reviewer 2 Report

The authors reply to my comments, therefore for me in this version the manuscript can be published.